# N2N Learning: Network to Network Compression via Policy Gradient Reinforcement Learning

**Anubhav Ashok**
Robotics Institute
Carnegie Mellon University
bhav@cmu.edu

**Nicholas Rhinehart**
Robotics Institute
Carnegie Mellon University
nrhineha@cs.cmu.edu

**Fares Beainy**
Volvo Construction Equipment
Volvo Group
fares.beainy@volvo.com

**Kris M. Kitani**
Robotics Institute
Carnegie Mellon University
kkitani@cs.cmu.edu

## Abstract

While wider and deeper neural network architectures continue to advance the state-of-the-art for many computer vision tasks, real-world adoption of these networks is impeded by hardware and speed constraints. Conventional model compression methods attempt to address this problem by modifying the architecture manually or using pre-defined heuristics. Since the space of all reduced architectures is very large, modifying the architecture of a deep neural network in this way is a difficult task. In this paper, we tackle this issue by introducing a principled method for *learning* reduced network architectures in a data-driven way using reinforcement learning. Our approach takes a larger 'teacher' network as input and outputs a compressed 'student' network derived from the 'teacher' network. In the first stage of our method, a recurrent policy network aggressively removes layers from the large 'teacher' model. In the second stage, another recurrent policy network carefully reduces the size of each remaining layer. The resulting network is then evaluated to obtain a reward – a score based on the accuracy and compression of the network. Our approach uses this reward signal with policy gradients to train the policies to find a locally optimal student network. Our experiments show that we can achieve compression rates of more than $10\times$ for models such as ResNet-34 while maintaining similar performance to the input 'teacher' network. We also present a valuable transfer learning result which shows that policies which are pre-trained on smaller 'teacher' networks can be used to rapidly speed up training on larger 'teacher' networks.

## 1 Introduction

While carefully hand-designed deep convolutional networks continue to increase in size and in performance, they also require significant power, memory and computational resources, often to the point of prohibiting their deployment on smaller devices. As a result, researchers have developed model compression techniques based on Knowledge Distillation to compress a large (teacher) network to a smaller (student) network using various training techniques (e.g., soft output matching, hint layer matching, uncertainty modeling). Unfortunately, state-of-the-art knowledge distillation methods share a common feature: they require carefully *hand-designed* architectures for the student model. Hand-designing networks is a tedious sequential process, often loosely guided by a sequence of trial-and-error based decisions to identify a smaller network architecture. This process makes it very difficult to know if the resulting network is optimal. Clearly, there is a need to develop more principled methods of identifying optimal student architectures.

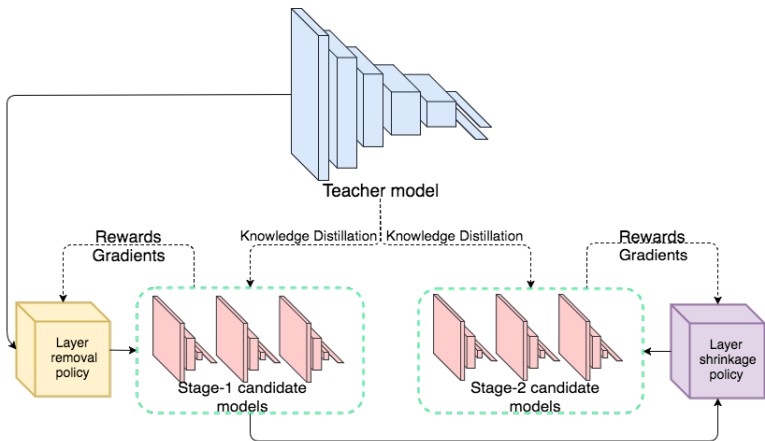

Figure 1: Layer Removal Policy removes layers of Teacher network architecture (stage-1 candidates) then Layer Shrinkage Policy reduces parameters (stage-2 candidates).

Towards a more principled approach to network architecture compression, we present a reinforcement learning approach to *identify a compressed high-performance architecture (student) given knowledge distilled from a larger high-performing model (teacher)*. We make a key conceptual assumption that formulates the sequential process of converting a teacher network to a student network as a Markov Decision Process (MDP). Under this model, a state $s$ represents the network architecture. Clearly, the domain of the state $\mathcal{S}$ is very large since it contains every possible reduced architecture of the teacher network. A deterministic transition in this state space, $T(s'|s, a)$, is determined by selecting the action $a$, e.g., removing a convolutional filter or reducing the size of a fully connected layer. Each action will transform one architecture $s$ to another architecture $s'$. Under the MDP, the strategy for selecting an action given a certain state is represented by the policy $\pi(a|s)$, which stochastically maps a state to an action. The process of reinforcement learning is used to learn an optimal policy based on a reward function $r(s)$ defined over the state space. In our work, we define the reward function based on the *accuracy* and the *compression rate* of the specified architecture $s$.

A straightforward application of reinforcement learning to this problem can be very slow depending on the definition of the action space. For example, an action could be defined as removing a single filter from every layer of a convolutional neural network. Since the search space is exponential in the size of the action space and sequence length, it certainly does not scale to modern networks that have hundreds of layers.

Our proposed approach addresses the problem of scalability in part, by introducing a two-stage action selection mechanism which first selects a macro-scale "layer removal" action, followed by a micro-scale "layer shrinkage" action. In this way we enable our reinforcement learning process to efficiently explore the space of reduced networks. Each network architecture that is generated by our policy is then trained with Knowledge Distillation (Hinton et al., 2015). Figure 1 illustrates our proposed approach.

To the best of our knowledge, this is the first paper to provide a principled approach to the task of network compression, where the architecture of the student network is obtained via reinforcement learning. To facilitate reinforcement learning, we propose a reward function that encodes both the compression rate and the accuracy of the student model. In particular, we propose a novel formulation of the compression reward term based on a relaxation of a constrained optimization problem, which encodes the hardware-based computational budget items in the form of linear constraints.

We demonstrate the effectiveness of our approach over several network architectures and several visual learning tasks of varying difficulty (MNIST, SVHN, CIFAR-10, CIFAR-100, Caltech-256). We also demonstrate that the compression policies exhibit generalization across networks with similar architectures. In particular, we use a policy trained on a ResNet-18 model on a ResNet-34 model and show that it greatly accelerates the reinforcement learning process.

## 2    RELATED WORK

We first discuss methods for compressing models to a manually designed network (pruning and distillation). Towards automation, we discuss methods for automatically constructing high-performance networks, orthogonal to the task of compression.

**Pruning:**    Pruning-based methods preserve the weights that matter most and remove the redundant weights LeCun et al. (1989), Hassibi et al. (1993), Srinivas & Babu (2015), Han et al. (2015b), Han et al. (2015a), Mariet & Sra (2015), Anwar et al. (2015), Guo et al. (2016). While pruning-based approaches typically operate on the weights of the teacher model, our approach operates on a much larger search space over both model weights and model architecture. Additionally, our method offers greater flexibility as it allows the enforcement of memory, inference time, power, or other hardware constraints. This allows our approach to find the optimal architecture for the given dataset and constraints instead of being limited to that of the original model.

**Knowledge Distillation:**    Knowledge distillation is the task of training a smaller network (a "student") to mimic a "teacher" network, performing comparably to the input network (a "teacher") Bucilu et al. (2006), Ba & Caruana (2014), Hinton et al. (2015), Romero et al. (2014), Urban et al. (2016). The work of Hinton et al. (2015) generalized this idea by training the student to learn from both the teacher and from the training data, demonstrating that this approach outperforms models trained using only training data. In Romero et al. (2014), the approach uses Knowledge Distillation with an intermediate hint layer to train a thinner but deeper student network containing fewer parameters to outperform the teacher network. In previous Knowledge Distillation approaches, the networks are hand designed, possibly after many rounds of trial-and-error. In this paper, we train a policy to learn the optimal student architecture, instead of hand-designing one. In a sense, we *automate Knowledge Distillation*, employing the distillation method of Ba & Caruana (2014) as a component of our learning process. In the experiments section we show that our learned architectures outperform those described in Romero et al. (2014) and Hinton et al. (2015).

**Architecture Search:**    There has been much work on exploring the design space of neural networks Saxe et al. (2011), Zoph & Le (2016), Baker et al. (2016), Ludermir et al. (2006), Miikku-lainen et al. (2017), Real et al. (2017), Snoek et al. (2012), Snoek et al. (2015), Stanley & Miikku-lainen (2002), Jozefowicz et al. (2015), Murdock et al. (2016), Feng & Darrell (2015), Warde-Farley et al. (2014), Iandola et al. (2016). The principal aim of previous work in architecture search has been to build models that maximize performance on a given dataset. On the other hand, our goal is to find a compressed architecture while maintaining reasonable performance on a given dataset. Our approach also differs from existing architecture search method since we use the teacher model as the search space for our architecture instead of constructing networks from scratch. Current methods that construct networks from scratch either operate on a very large search space, making it computationally expensive Zoph & Le (2016), Real et al. (2017), Miikkulainen et al. (2017), Jozefowicz et al. (2015) or operate on a highly restricted search space Baker et al. (2016), Snoek et al. (2015). Our approach instead leverages the idea that since the teacher model is able to achieve high accuracy on the dataset, it already contains the components required to solve the task well and therefore is a suitable search space for the compressed architecture.

## 3    APPROACH

Our goal is to learn an optimal compression strategy (policy) via reinforcement learning, that takes a Teacher network as input and systematically reduces it to output a small Student network.

### 3.1    MARKOV DECISION PROCESS

We formulate the sequential process of finding a reduced architecture as a sequential decision making problem. The decision process is modeled as a Markov Decision Process (MDP). Formally, the MDP is defined as the tuple $\mathcal{M} = \{\mathcal{S}, \mathcal{A}, T, r, \gamma\}$.

**States:** $\mathcal{S}$ is the state space, a finite set consisting of all possible reduced network architectures that can be derived from the Teacher model. For example, a VGG network (Simonyan & Zisserman,

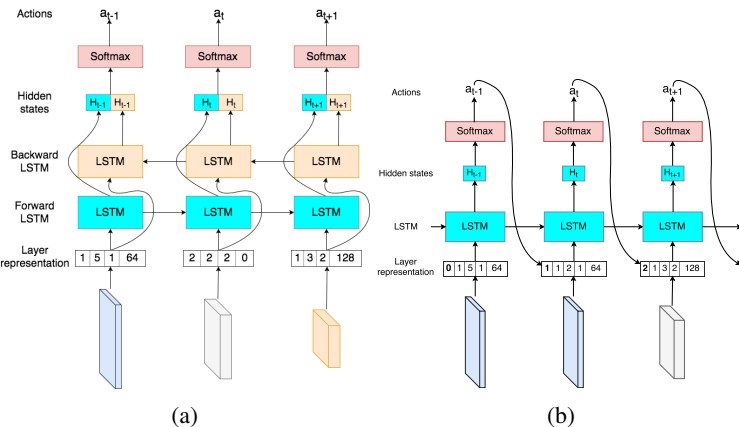

Figure 2: **a)** Layer removal policy network, **b)** Layer shrinkage policy network

2014) represents the state $s \in \mathcal{S}$ (the initial state) and by removing one convolutional filter from the first layer we obtain a new network architecture $s'$.

**Actions:** $\mathcal{A}$ is a finite set of actions that can transform one network architecture into another network architecture. In our approach there are two classes of action types: layer removal actions and layer parameter reduction actions. The definition of these actions are further described in Section 3.2.1 and 3.2.2.

**Transition Function:** $T : \mathcal{S} \times \mathcal{A} \to \mathcal{S}$ is the state transition dynamic. Here, $T$ is deterministic since an action $a$ always transforms a network architecture $s$ to the resulting network architecture $s'$ with probability one.

**Discount Factor:** $\gamma$ is the discount factor. We use $\gamma = 1$ so that all rewards contribute equally to the final return.

**Reward:** $r : \mathcal{S} \to \mathbb{R}$ is the reward function. The rewards of network architecture $r(s)$ can be interpreted to be a score associated with a given network architecture $s$. Note that we define the reward to be 0 for intermediate states, which represent "incomplete" networks, and only compute a non-trivial reward for the final state. The reward function is described in detail in Section 3.4.

## 3.2 STUDENT-TEACHER REINFORCEMENT LEARNING

Under this MDP, the task of reinforcement learning is to learn an optimal policy $\pi : \mathcal{S} \to \mathcal{A}$, such that it maximizes the expected total reward, with the total reward given by:

$$R(\vec{s}) = \sum_{i=0}^{L=|\vec{s}|} r(s_i) = r(s_L). \tag{1}$$

We take a policy gradient reinforcement learning approach and iteratively update the policy based on sampled estimates of the reward. The design of the action space is critical for allowing the policy gradient method to effectively search the state space. If the actions are selected to be very incremental, a long sequence of actions would be needed to make a significant change to the network architecture, making credit assignment difficult. To address this issue, we propose a two stage reinforcement learning procedure. In the first stage a policy selects a sequence of actions deciding whether to keep or remove each layer of the teacher architecture. In the second stage, a different policy selects a sequence of discrete actions corresponding to the magnitude by which to attenuate configuration variables of each remaining layer. In this way, we are able to efficiently explore the state space to find the optimal student network.

---

**Algorithm 1** Student-Teacher Reinforcement Learning

---

1: **procedure** STUDENT-TEACHER RL($\mathcal{S}, \mathcal{A}, T, r, \gamma$)
2:     $s_0 \leftarrow$ Teacher
3:     **for** $i = 1$ to $N_1$ **do**                                                    ▷ Layer removal
4:         **for** $t = 1$ to $L_1$ **do**
5:             $a_t \sim \pi_{\text{remove}}(s_{t-1}; \theta_{\text{remove}, i-1})$
6:             $s_t \leftarrow T(s_{t-1}, a_t)$
7:         **end for**
8:         $R \leftarrow r(s_{L_1})$
9:         $\theta_{\text{remove}, i} \leftarrow \nabla_{\theta_{\text{remove}, i-1}} J(\theta_{\text{remove}, i-1})$               ▷ (Eq. 2)
10:     **end for**
11:     $s_0 \leftarrow$ Stage-1 Candidate
12:     **for** $i = 1$ to $N_2$ **do**                                                  ▷ Layer shrinkage
13:         **for** $t = 1$ to $L_2$ **do**
14:             $a_t \sim \pi_{\text{shrink}}(s_{t-1}; \theta_{\text{shrink}, i-1})$
15:             $s_t \leftarrow T(s_{t-1}, a_t)$
16:         **end for**
17:         $R \leftarrow r(s_{L_2})$
18:         $\theta_{\text{shrink}, i} \leftarrow \nabla_{\theta_{\text{shrink}, i-1}} J(\theta_{\text{shrink}, i-1})$           ▷ (Eq. 2)
19:     **end for**
20:     **Output:** Compressed model
21: **end procedure**

---

A sketch of the algorithm is given in Algorithm 3.2. For both layer removal and shrinkage policies, we repeatedly sample architectures and update the policies based on the reward achieved by the architectures. We now describe the details of the two stages of student-teacher reinforcement learning.

### 3.2.1 LAYER REMOVAL

In the layer removal stage, actions $a_t$ correspond to the binary decision to keep or remove a layer. The length of the trajectory for layer removal is $T = L$, the number of layers in the network. At each step $t$ of layer removal, the Bidirectional LSTM policy (See Figure 2a) observes the hidden states, $h_{t-1}, h_{t+1}$, as well as information $x_t$ about the current layer: $\pi_{\text{remove}}(a_t | h_{t-1}, h_{t+1}, x_t)$. Information about the current layer $l$ is given as

$$x_t = (l, k, s, p, n, s_{\text{start}}, s_{\text{end}}),$$

where $l$ is the layer type, $k$ kernel size, $s$ stride, $p$ padding and $n$ number of outputs (filters or connections). To model more complex architectures, such as ResNet, $s_{\text{start}}$ and $s_{\text{end}}$ are used to inform the policy network about skip connections. For a layer inside a block containing a skip connection, $s_{\text{start}}$ is the number of layers prior to which the skip connection began and $s_{\text{end}}$ is the number of layers remaining until the end of the block. Additionally it is to be noted that although actions are stochastically sampled from the outputs at each time step, the hidden states that are passed on serve as a sufficient statistic for $x_0, a_0 ... x_{t-1}, a_{t-1}$ (Wierstra et al., 2010).

### 3.2.2 LAYER SHRINKAGE

The length of the trajectory for layer shrinkage is $T = \sum_{l=1}^{L} H_l$, where $H$ is the number of configuration variables for each layer. At each step $t$ of layer shrinkage, the policy observes the hidden state $h_{t-1}$, the previously sampled action $a_{t-1}$ and current layer information $x_t$: $\pi_{\text{shrink}}(a_t | a_{t-1}, h_{t-1}, x_t)$. The parameterization of $x_t$ is similar to layer removal except that the previous action is appended to the representation in an autoregressive manner (See Figure 2b). The action space for layer shrinkage is defined as $a_t \in [0.1, 0.2, \ldots, 1]$ (each action corresponds to how much to shrink a layer parameter) and an action is produced for each configurable variable for each layer. Examples include kernel size, padding, and number of output filters or connections.

### 3.3 REWARD FUNCTION

The design of the reward function plays a critical role in learning the policies. A poorly designed reward that provides no discrimination between good and bad student architectures prevents policies

from learning the trade-offs in architecture space. The objective of model compression is to maximize compression while maintaining a high accuracy. Since there is no benefit in producing highly compressed models which have bad performance, we want to provide a harsher penalty for a model with high compression + low accuracy than one with low compression + high accuracy. Furthermore we would also like to define a general reward function that does not depend on dataset/model specific hyperparameters. Additional discussion on the design of the reward function is provided in the appendix.

In our approach, we define the reward function as follows:

$$R = R_c \cdot R_a$$
$$= C(2 - C) \cdot \frac{A}{A_{\text{teacher}}}$$

Where $C$ is the relative compression ratio of the student model, $A$ is the validation accuracy of the student model and $A_{\text{teacher}}$ is the validation accuracy of the teacher model provided defined as a constant. $R_c$ and $R_a$ refer to the compression and accuracy reward respectively. We compute the reward as a product of the compression and accuracy reward since we want the reward to scale with both quantities dependently. The compression reward, $R_c = C(2 - C)$, is computed using a non-linear function that biases the policy towards producing models that maintain accuracy while optimizing for compression. The relative compression $C \in [0, 1)$ is defined in terms of the ratio of trainable parameters of each model: $C = 1 - \frac{\#\text{params(student)}}{\#\text{params(teacher)}}$. It is noted here that other compression methods that use quantization or coding define compression ratio in terms of number of bits instead of parameters. The accuracy reward, $R_a$, is defined with respect to the teacher model as $R_a = \frac{A}{A_{\text{teacher}}}$, where $A \in [0, 1]$ refers to the validation accuracy of the student model and $A_{\text{teacher}}$ refers to the validation accuracy of the teacher model. We note that both accuracy and compression rewards are normalized with respect to the teacher and thus do not require additional hyperparameters to perform task-specific weighting. Lastly, it is possible that the policies may produce degenerate architectures in such cases, a reward if -1 is assigned (details in appendix).

### 3.3.1 Constraints as Rewards

Our approach allows us to incorporate pre-defined hardware or resource budget constraints by rewarding architectures that meet the constraints and discouraging those that do not. Formally, our constrained optimization problem is

$$\max E_{a_{1:T}}[R]$$
$$\text{subject to } Ax \leq b,$$

where $A$ and $b$ form our constraints, and $x$ is vector of constrained variables. We relax these hard constraints by redefining our reward function as:

$$R = \begin{cases} R_a \cdot R_c & \text{if } Ax \leq b \\ -1 & \text{otherwise.} \end{cases}$$

The introduction of the non-smooth penalty may result in a reduced exploration of the search space and hence convergence to a worse local minimum. To encourage early exploration gradually incorporate constraints over time:

$$R = \begin{cases} R_a \cdot R_c & \text{if } Ax \leq b \\ \epsilon_t(R_a \cdot R_c + 1) - 1 & \text{otherwise,} \end{cases}$$

where $\epsilon_t \in [0, 1]$ monotonically decreases with $t$ and $\epsilon_0 = 1$. As it is possible to incorporate a variety of constraints such as memory, time, power, accuracy, label-wise accuracy, our method is flexible enough to produce models practically viable in a diversity of settings. This is in contrast to conventional model compression techniques which require many manual repetitions of the algorithm in order to find networks that meet the constraints as well as optimally balance the accuracy-size tradeoff.

### 3.4 Optimization

We now describe the optimization procedure for each our stochastic policies, $\pi_{\text{remove}}$ and $\pi_{\text{shrink}}$. The procedure is the same for each policy, thus we use $\pi$ in what follows. Each policy network is parameterized by its own $\theta$.

Our objective function is the expected reward over all sequences of actions $a_{1:T}$, i.e.:

$$J(\theta) = E_{a_{1:T} \sim P_\theta}(R)$$

We use the REINFORCE policy gradient algorithm from Williams (1992) to train both of our policy networks.

$$\nabla_\theta J(\theta) = \nabla_\theta E_{a_{1:T} \sim P_\theta}(R)$$

$$= \sum_{t=1}^{T} E_{a_{1:T} \sim P_\theta}[\nabla_\theta \log P_\theta(a_t | a_{1:(t-1)}) R]$$

$$\approx \frac{1}{m} \sum_{k=1}^{m} \sum_{t=1}^{T} [\nabla_\theta \log P_\theta(a_t | h_t) R_k]$$

where $m$ is the number of rollouts for a single gradient update, $T$ is the length of the trajectory, $P_\theta(a_t | h_t)$ is the probability of selecting action $a_t$ given the hidden state $h_t$, generated by the current stochastic policy parameterized by $\theta$ and $R_k$ is the reward of the $k^{\text{th}}$ rollout.

The above is an unbiased estimate of our gradient, but has high variance. A common trick is to use a state-independent baseline function to reduce the variance:

$$\nabla_\theta J(\theta) \approx \frac{1}{m} \sum_{k=1}^{m} \sum_{t=1}^{T} [\nabla_\theta \log P_\theta(a_t | h_t)(R_k - b)] \tag{2}$$

We use an exponential moving average of the previous rewards as the baseline $b$. An Actor-Critic policy was also tested. While there was a minor improvement in stability, it failed to explore as effectively in some cases, resulting in a locally optimal solution. Details are in the appendix.

### 3.5 Knowledge distillation

Student models are trained using data labelled by a teacher model. Instead of using hard labels, we use the un-normalized log probability values (the logits) of the teacher model. Training using the logits helps to incorporate *dark knowledge* (Hinton et al., 2015) that regularizes students by placing emphasis on the relationships learned by the teacher model across all of the outputs.

As in Ba & Caruana (2014), the student is trained to minimize the mean $L_2$ loss on the training data $\left\{(x^i, z^i)\right\}_{i=1}^{N}$. Where $z^i$ are the logits of the teacher model.

$$\mathcal{L}_{\text{KD}}(f(x; W), z) = \frac{1}{N} \sum_i ||f(x^{(i)}; W) - z^{(i)}||_2^2$$

where W represents the weights of the student network and $f(x^{(i)}; W)$ is the model prediction on the $i^{th}$ training data sample.

Final student models were trained to convergence with hard and soft labels using the following loss function.

$$\mathcal{L}(\mathcal{W}) = \mathcal{L}_{\text{hard}}(f(x; W), y_{\text{true}}) + \lambda * \mathcal{L}_{\text{KD}}(f(x; W), z)$$

Where $\mathcal{L}_{\text{hard}}$ is the loss function used for training with hard labels (in our case cross-entropy) and $y_{\text{true}}$ are the ground truth labels.

## 4 Experiments

In the following experiments, we first show that our method is able to find highly compressed student architectures with high performance on multiple datasets and teacher architectures, often exceeding performance of the teacher model. We compare the results obtained to current baseline methods of model compression, showing competitive performance. Then we demonstrate the viability of our method in highly resource constrained conditions by running experiments with strong model size constraints. Finally, we show that it is possible to rapidly speed up training when using larger teacher models by reusing policies that are pretrained on smaller teacher models.

Table 1: Summary of Compression results.

| MNIST | | | | | |
| --- | --- | --- | --- | --- | --- |
| Architecture | | Acc. | #Params | $\Delta$ Acc. | Compr. |
| VGG-13 | Teacher | 99.54% | 9.4M | — | — |
| | Student (Stage1) | 99.55% | 73K | +0.01% | 127x |
| CIFAR-10 | | | | | |
| VGG-19 | Teacher | 91.97% | 20.2M | — | — |
| | Student (Stage1) | 92.05% | 1.7M | +0.08% | 11.8x |
| | Student (Stage1+Stage2) | 91.64% | 984K | -0.33% | 20.53x |
| ResNet-18 | Teacher | 92.01% | 11.17M | — | — |
| | Student (Stage1) | 91.97% | 2.12M | -0.04% | 5.26x |
| | Student (Stage1+Stage2) | 91.81% | 1.00M | -0.2% | 11.10x |
| ResNet-34 | Teacher | 92.05% | 21.28M | — | — |
| | Student (Stage1) | 93.54% | 3.87M | +1.49% | 5.5x |
| | Student (Stage1+Stage2) | 92.35% | 2.07M | +0.30% | 10.2x |
| SVHN | | | | | |
| ResNet-18 | Teacher | 95.24% | 11.17M | — | — |
| | Student (Stage1) | 95.66% | 2.24M | +0.42% | 4.97x |
| | Student (Stage1+Stage2) | 95.38% | 564K | +0.18% | 19.8x |
| CIFAR-100 | | | | | |
| ResNet-18 | Teacher | 72.22% | 11.22M | — | — |
| | Student (Stage1) | 69.64% | 4.76M | -2.58% | 2.35x |
| | Student (Stage1+Stage2) | 68.01% | 2.42M | -4.21% | 4.64x |
| ResNet-34 | Teacher | 72.86% | 21.33M | — | — |
| | Student (Stage1) | 70.11% | 4.25M | -2.75% | 5.02x |
| Caltech256 | | | | | |
| ResNet-18 | Teacher | 47.65% | 11.31M | — | — |
| | Student (Stage1) | 44.71% | 3.62M | -2.94% | 3.12x |
| | Student (Stage1+Stage2) | 44.63% | 2.45M | -3.02% | 4.61x |
| ImageNet32x32 | | | | | |
| ResNet-34 | Teacher | 30.87% | 21.79M | — | — |
| | Student (Stage1) | 30.22% | 3.34M | -0.65% | 6.51x |

## 4.1 DATASETS

**MNIST** The MNIST (LeCun et al., 1998) dataset consists of $28 \times 28$ pixel grey-scale images depicting handwritten digits. We use the standard 60,000 training images and 10,000 test images for experiments. Although MNIST is easily solved with smaller networks, we used a high capacity models (e.g., VGG-13) to show that the policies learned by our approach are able to effectively and aggressively remove redundancies from large network architectures.

**CIFAR-10** The CIFAR-10 (Krizhevsky & Hinton, 2009) dataset consists of 10 classes of objects and is divided into 50,000 train and 10,000 test images (32x32 pixels). This dataset provides an incremental level of difficulty over the MNIST dataset, using multi-channel inputs to perform model compression.

**SVHN** The Street View House Numbers (Netzer et al., 2011) dataset contains 3232 colored digit images with 73257 digits for training, 26032 digits for testing. This dataset is slightly larger that CIFAR-10 and allows us to observe the performance on a wider breadth of visual tasks.

**CIFAR-100** To further test the robustness of our approach, we evaluated it on the CIFAR-100 dataset. CIFAR-100 is a harder dataset with 100 classes instead of 10, but the same amount of data, 50,000 train and 10,000 test images (32x32). Since there is less data per class, there is a

steeper size-accuracy tradeoff. We show that our approach is able to produce solid results despite these limitations.

**Caltech-256** To test the effectiveness of our approach in circumstances where data is *sparse*, we run experiments on the Caltech-256 dataset (Griffin et al., 2007). This dataset contains more classes and less data per class than CIFAR-100, containing 256 classes and a total of 30607 images (224x224). We trained the networks from scratch instead of using pretraining in order to standardize our comparisons across datasets.

**ImageNet32x32** To test the efficiency of our approach, an experiment was conducted on a large scale dataset, ImageNet32x32 (Chrabaszcz et al., 2017). This dataset contains the same training/validation splits as the original ImageNet (Krizhevsky et al., 2012) dataset. It consists of 1.28 million training images and 50,000 validation images with 1000 object classes. However, unlike the original ImageNet dataset which uses 224x224 RGB images, ImageNet32x32 uses 32x32 RGB images, which reduces training time while increasing the difficulty of the task.

## 4.2 TRAINING DETAILS

In the following experiments, student models were trained as described in Section 3.5. We observed heuristically that 5 epochs was sufficient to compare performance.

The layer removal and layer shrinkage policy networks were trained using the Adam optimizer with a learning rate of 0.003 and 0.01 respectively. Both recurrent policy networks were trained using the REINFORCE algorithm (batch size=5) with standard backpropagation through time. A grid search was done to determine the ideal learning rate and batch size (details in appendix).

## 4.3 COMPRESSION EXPERIMENTS

In this section we evaluate the ability of our approach to learn policies to find compressed architectures without any constraints. In the following experiments, we expect that the policies learned by our approach will initially start out as random and eventually tend towards an optimal size-accuracy trade-off which results in a higher reward. Definitions of architectures are available in the appendix.

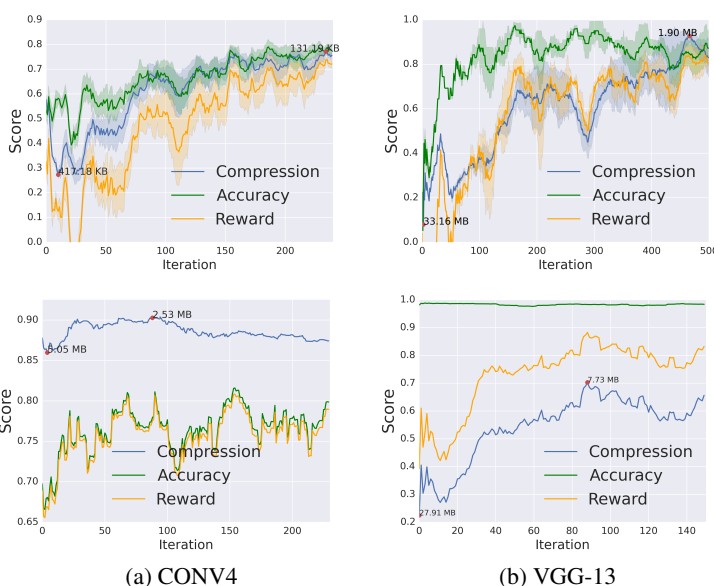

Figure 3: Student learning on **MNIST**. Reward, Accuracy, Compression vs Iteration (**Top**: Stage 1, **Bottom**: Stage 2)

**MNIST** To evaluate the compression performance we use (1) a **Conv4** network consisting of 4 convolutional layers and (2) a high capacity **VGG-13** network.

Figure 3 shows the results of our compression approach for each teacher network. The lines represent the compression (blue), accuracy (green) and reward (orange). The y-axis represents the score of those quantities, between 0 and 1. The x-axis is the iteration number. We also highlight the largest and smallest models with red circles to give a sense of the magnitude of compression. This experiment appears to confirm our original expectation that the policies would improve over time.

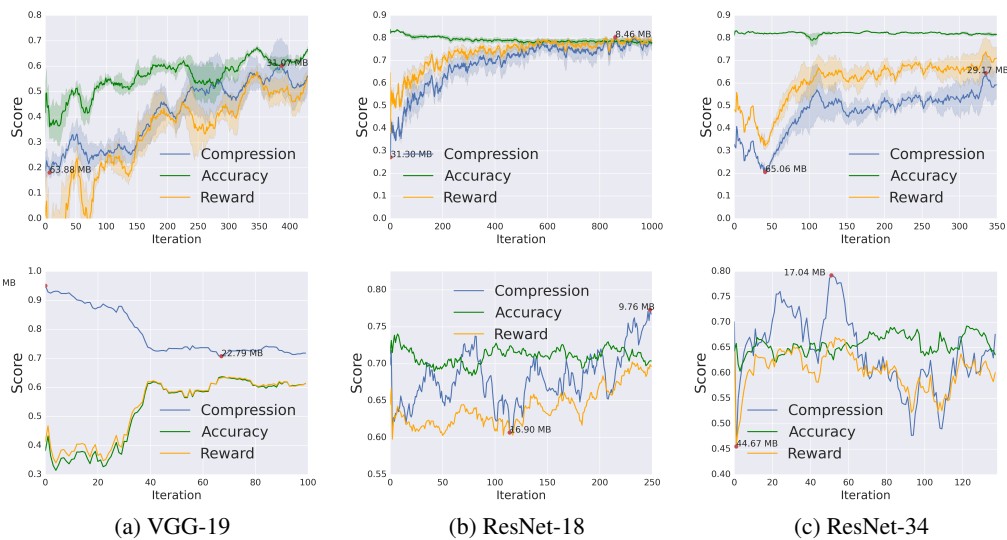

(a) VGG-19     (b) ResNet-18     (c) ResNet-34

Figure 4: Student learning on **CIFAR-10**. Reward, Accuracy, Compression vs Iteration (**Top**: Stage 1, **Bottom**: Stage 2)

**CIFAR-10** On the CIFAR-10 dataset we ran experiments using the following teacher networks: (1) **VGG-19**, (2) **ResNet-18** and (3) **ResNet-34** networks. The experimental results are shown in Figure 4. It is interesting to note that on CIFAR-10, our learned student networks perform almost as well or better the teacher networks despite a 10x compression rate.

**SVHN** On the SVHN dataset, we ran experiments using **ResNet-18** network as the teacher model. We observed that the reward and compression steadily increased while the accuracy remained stable, confirming similar results to that of CIFAR-10. This is a promising indication that our approach works for a breadth of tasks and isn't dataset specific. Results are in the appendix.

**CIFAR-100** We also verified our approach on a harder dataset, CIFAR-100 to show how our approach performs with less data per class (Figure 5). Considering the largely reduced number of parameters, the compressed network achieves reasonably high accuracy. A notable aspect of many of the final compressed models is that ReLU layers within residual blocks were removed. Another interesting result is that the compressed ResNet-34 student model outperforms the ResNet-18 model despite having fewer parameters. This can likely be explained by the increased number of residual blocks in the ResNet-34 model.

**Caltech-256** The Caltech-256 experiments (appendix) show the performance of our approach when training data is scarce. We would like to verify that our approach does not overly compress the network by overfitting to the small number of training examples. As with the other experiments, the policies appears to learn to maximize reward over time, although the positive trend is not as pronounced due to the lack of training data. This is expected since less data means the reward signal is less robust to sources of noise, which in turn affects training of the policy.

**ImageNet32x32** We conducted an experiment on the ImageNet32x32 dataset to test the performance of our approach on a large scale dataset. Due to the increased difficulty of this dataset, the teacher model (ResNet-34) achieved a top-1 accuracy of 30.87% after training for 40 epochs. Despite the difficulty of the dataset, our approach was still able to find a compressed model with similar performance (-0.65% drop). The runtime for 100 iterations of the layer removal policy on a ResNet-34 teacher and a batch size of 3 was approximately 272 hours. More details regarding the runtime can be found in Section 12.

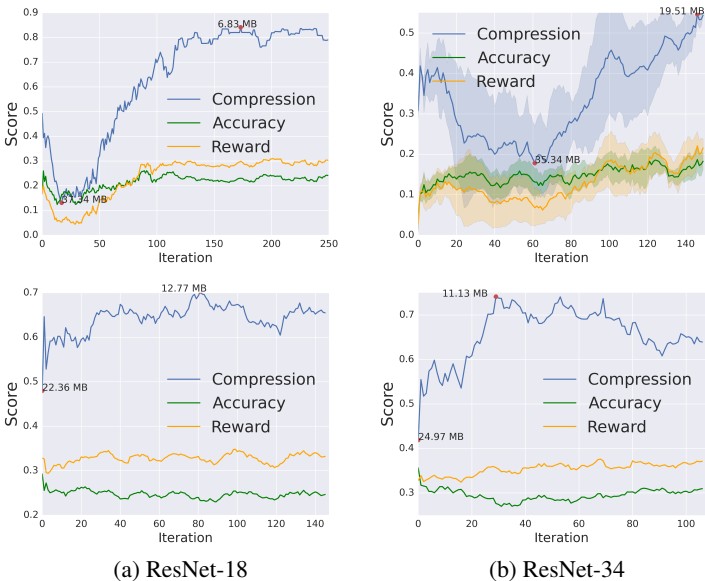

(a) ResNet-18                          (b) ResNet-34

Figure 5: Student learning on **CIFAR-100**.

## 4.4 BASELINES

We compare the performance of our approach to current model compression methods, namely pruning and Knowledge Distillation (with hand-designed model). We note here that compression rate is defined as the ratio of number of parameters instead of number of bits, which some other compression methods (quantization, coding) use. To provide a fair comparison with our method, the same trained teacher models used in our method were used.

### 4.4.1 PRUNING

Table 2: Pruning (Baseline)

| Model | Acc. | #Params | Compr. | Δ Acc. |
|---|---|---|---|---|
| Teacher (MNIST/VGG-13) | 99.54% | 9.4M | — | — |
| Pruning | 99.12% | 162K | 58x | -0.42% |
| Ours | **99.55%** | **73K** | **127x** | **+0.01%** |
| Teacher (CIFAR-10/VGG-19) | 91.97% | 20.2M | — | — |
| Pruning | 91.06% | 2.3M | 8.7x | -0.91% |
| Ours | **92.05%** | **1.7M** | **11.8x** | **+0.08%** |

We compare our method to pruning, which is a model compression approach that operates directly on the weight space of a network, removing redundant weights or filters. We perform pruning based on Molchanov et al. (2016), which removes filters using a greedy criteria based approach and then finetunes the network. With pruning, the performance of the final model can vary depending on the degree to which it was pruned. To ensure a fair comparison, we stop pruning when 1. accuracy drops below 1% of the student model obtained by our method or 2. the number of parameters is less than our method. Pruning is done 5 times to control for variance and the best performing model is reported.

The results of this experiment, reported in Table 2, show that while the pruned models show good compression rates, our approach outperforms this baseline on both datasets. These results could indicate that operating on the architecture space of the model might result in more consistent results than using heuristics to operate on the weight space directly.

### 4.4.2 Knowledge Distillation

Table 3: Knowledge distillation with hand designed models (Baseline)

| Model | Acc. | #Params | Compr. | Δ Acc. |
|---|---|---|---|---|
| Teacher (SVHN/ResNet-18) | 95.24% | 11.17M | — | — |
| SqueezeNet1.1 | 89.34% | 727K | 15x | -5.90% |
| **Ours** | **95.38%** | **564K** | **19.8x** | **+0.18%** |
| Teacher (CIFAR-10/ResNet-18) | 92.01% | 11.17M | — | — |
| FitNet-4 | 91.33% | 1.2M | 9.3x | -0.63% |
| VGG-small | 83.93% | 1.06M | 10.5x | -8.08% |
| **Ours** | **91.81%** | **1.00M** | **11.0x** | **-0.20%** |

We also tested the validity of our hypothesis that hand designed models may not be optimal for Knowledge distillation. We compare models generated by our method to hand designed models that contain a similar number of parameters. We perform experiments with 3 hand designed model architectures, FitNet-4, SqueezeNet and a reduced network based on VGG, (VGG-small) which contains 10 layers. These networks were then trained to convergence with Knowledge Distillation on the CIFAR-10 dataset and the SVHN datasets.

For the implementation of FitNet-4 (17 layers), we used the same model architecture described in Mishkin & Matas (2015) with the ReLU activation and Xavier initialization. The paper reported a baseline accuracy of 90.63 when trained from scratch and 1.2 M parameters (Table 3 in Mishkin & Matas (2015)). For SqueezeNet, we implemented the 1.1 version described in Iandola et al. (2016), which contained 727K parameters after adapting it to CIFAR-10. We benchmarked VGG-small and FitNet on the CIFAR-10 dataset and SqueezeNet on the SVHN dataset in order to provide a fair comparison with our best models in terms of the number of parameters.

From the results reported in Table 3, we observe that our method performs better than the hand-designed models on both datasets despite containing fewer parameters. The CIFAR-10 results seem to indicate that model selection is an important factor in Knowledge Distillation. Our model and the FitNet-4 model both outperform the VGG-small model, further confirming our hypothesis that hand-designing models may not be the optimal approach for use with Knowledge Distillation.

### 4.5 Compression with Size Constraints

Table 4: Model Compression with Size Constraints

| Model | Acc. | #Params | Compr. | Constr. |
|---|---|---|---|---|
| Teacher (MNIST/VGG-13) | **99.54%** | 9.4M | 1x | N/A |
| Student (Stage 1 & 2) | **98.91**% | **17K** | **553x** | 20K |
| Teacher (CIFAR-10/VGG-19) | **91.97%** | 20.2M | 1x | N/A |
| Student (Stage 1 & 2) | **90.8**% | **573K** | **35x** | 1M |

While the experiments to this point used no explicit constraints, in this experiment, we add a size constraint in terms of the number of parameters via the reward function as in Section 3.3.1. We expect the optimization to be harder because the range of acceptable architectures is reduced.

Results are summarized in Table 4. These promising results suggest that the compression policies are able to produce sensible results despite being heavily constrained, thus demonstrating the viability of the approach in practice.

### 4.6 Transfer Learning

Naively applying our approach to a new teacher network means that the compression policies must be learned from scratch for each new problem. We would like to know if layer removal and shrinkage policy networks can be reused to accelerate compression for new teacher architectures. In the

Table 5: Transfer Learning Performance during first 10 iterations.

| | ResNet18 → ResNet34 | | | ResNet34→ ResNet18 | | | VGG11→ VGG19 | | |
|---|---|---|---|---|---|---|---|---|---|
| | Reward | Comp. | Acc. | Reward | Comp. | Acc. | Reward | Comp. | Acc. |
| Pre-trained | **0.81** | **78.1%** | 79.5% | **0.76** | **65.5%** | 82.3% | **0.52** | **46.0%** | **71.7%** |
| Scratch | 0.50 | 34.8% | **82.4%** | 0.53 | 39.7% | **82.8%** | -0.07 | 20.2% | 42.5 % |

following experiments, we train a policy on an initial teacher model and then apply it to another teacher model to test whether the policy has learned a general strategy for compressing a network. Since both a pretrained policy and a randomly initialized policy is expected to eventually converge to a locally optimal policy given enough iterations, we provide performance measures over the the first 10 policy update iterations.

Results are summarized in Table 5. The slight drop in accuracy (third subcolumn) in models produced by the pretrained policy is expected due to the tradeoff between compression and accuracy. However, the average reward (first subcolumn) is always higher when we use a pretrained policy. Note that in the VGG experiment, the reward is negative since the non-pretrained policy starts off by producing degenerate models. However, the pretrained policy starts off from a different initialization that does not.

This is an important result as it shows promising evidence that we can even transfer learned knowledge from a *smaller* model to a *larger* model, rapidly accelerating the policy search procedure on very deep networks.

## 5    CONCLUSION

We introduced a novel method for compressing neural networks. Our approach employs a two-stage layer removal and layer shrinkage procedure to learn how to compress large neural networks. By leveraging signals for accuracy and compression as supervision, our method efficiently learns to search the space of model architectures. We show that our method performs well over a variety of datasets and architectures. We also observe generalization capabilities of our method through transfer learning, allowing our procedure to be made even more efficient. Our method is also able to incorporate other practical constraints, such as power or inference time, thus showing potential for application in a real world setting.

## ACKNOWLEDGEMENTS

This work was sponsored in part by IARPA (D17PC00340).

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

APPENDIX

## 6 ACTOR-CRITIC

Policy gradient based Actor-Critic algorithms have been shown to improve the stability of the policy search. This is achieved by replacing the baseline with a learned estimate of the value function at each time step.

Formally, with vanilla REINFORCE we have,

$$\nabla_\theta J(\theta) \approx \frac{1}{m} \sum_{k=1}^{m} \sum_{t=1}^{T} [\nabla_\theta \log P_\theta(a_t|h_t)(R_k - b_k)]$$

In the Actor-Critic algorithm we replace $b_k$ with $V_k^\theta$, resulting in a new gradient estimate,

$$\nabla_\theta J(\theta) \approx \frac{1}{m} \sum_{k=1}^{m} \sum_{t=1}^{T} [\nabla_\theta \log P_\theta(a_t|h_t)(R_k - V_k^\theta)]$$

We implement the Critic network by adding an additional fully-connected layer that takes as input the hidden state of the LSTM and outputs a single scalar value. Figures 6-7 the results of the experiments performed.

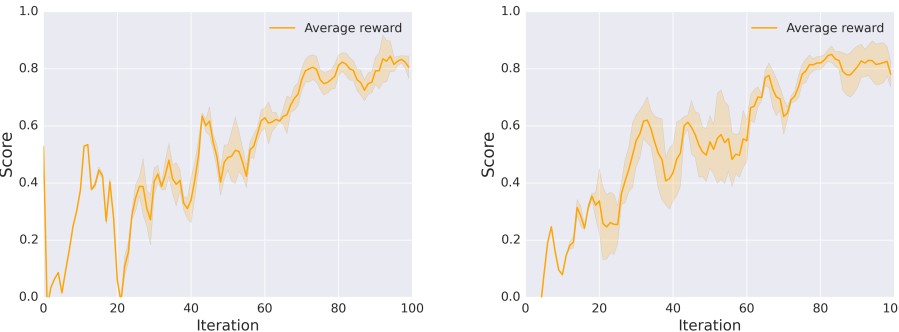

Figure 6: MNIST **Left:** Actor-critic **Right:** REINFORCE, averaged over 3 runs

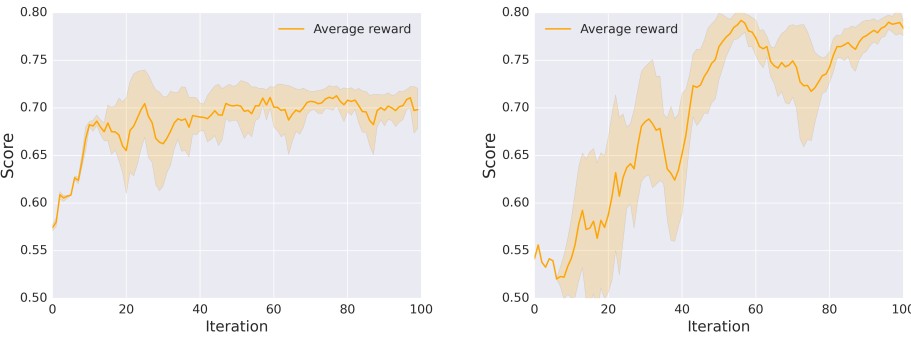

Figure 7: CIFAR-10 **Left:** Actor-critic **Right:** REINFORCE, averaged over 3 runs

For the MNIST dataset, our results show that there is a slight improvement in stability, although they both converge at a similar rate.

For the CIFAR-10 dataset, although the Actor-critic version was more stable, it did not perform as well as the vanilla REINFORCE algorithm.

## 7 LEARNING RATE AND BATCH SIZE

The learning rate and batch size were selected via a grid search. The following graphs show the rate of convergence for different learning rates and batch sizes.

### 7.1 LEARNING RATE

In order to determine the learning rate, we performed a grid search over 0.03, 0.003, 0.0003. We performed this grid search on the MNIST dataset using the VGG-13 network to save time. For the stage-1 policy, it was observed that lr=0.03 did not converge while lr=0.0003 converged too slowly. Thus we used lr=0.003 as the learning rate.

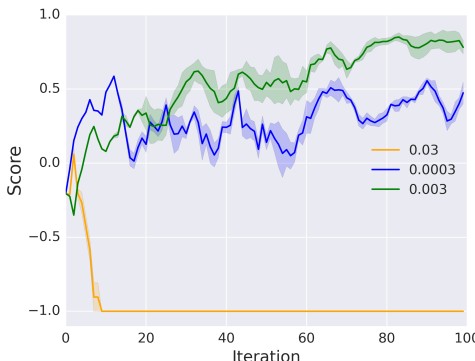

Figure 8: Average reward over 3 runs for various learning rates on the MNIST dataset

### 7.2 BATCH SIZE

Similarly we performed a grid search to determine the optimal batch size over 1, 5, 10. A batch size of 1 was too unstable while a batch size of 10 offered no substantial improvements to justify the additional computation. Thus we observed that a batch size of 5 worked the best.

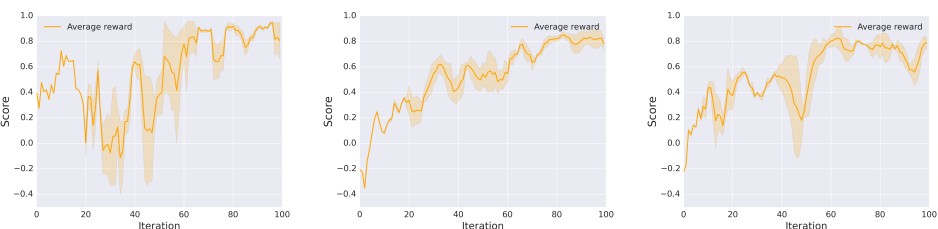

Figure 9: Average reward over 3 runs for batch sizes **Left:** 1, **Middle:** 5, **Right:** 10 on the MNIST dataset

## 8 TRANSFER LEARNING EXPERIMENTS

Below are the results of the transfer learning experiments, as observed, the pretrained policies start off with a high reward unlike the policies trained from scratch.

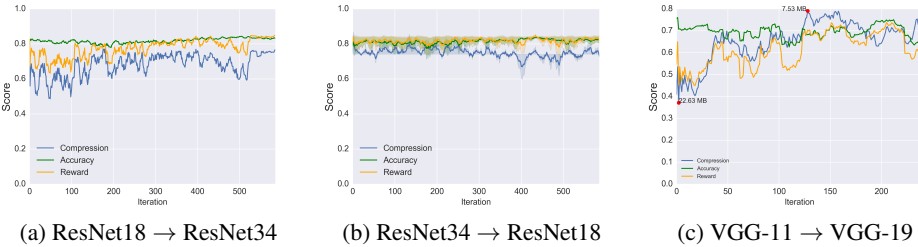

(a) ResNet18 → ResNet34    (b) ResNet34 → ResNet18    (c) VGG-11 → VGG-19

Figure 10: Transfer learning experiments

# 9 ADDITIONAL EXPERIMENTS

The following section contains results about additional compression experiments that were conducted.

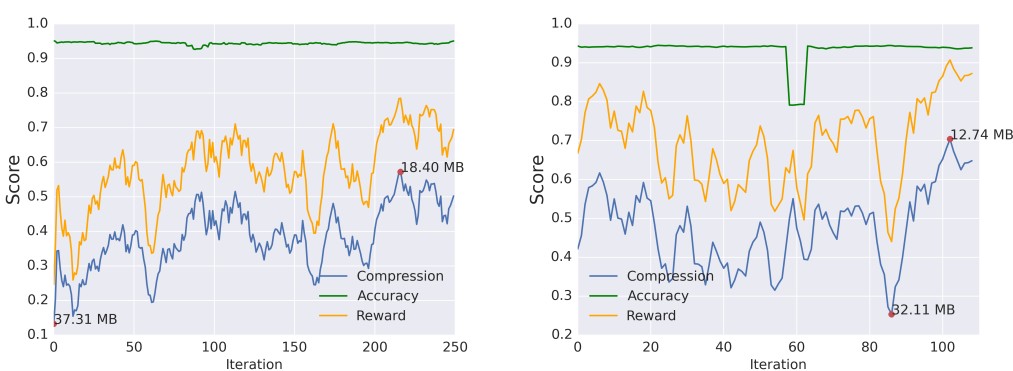

Figure 11: ResNet-18 experiments on **SVHN**, (**Left**: Stage 1, **Right**: Stage 2)

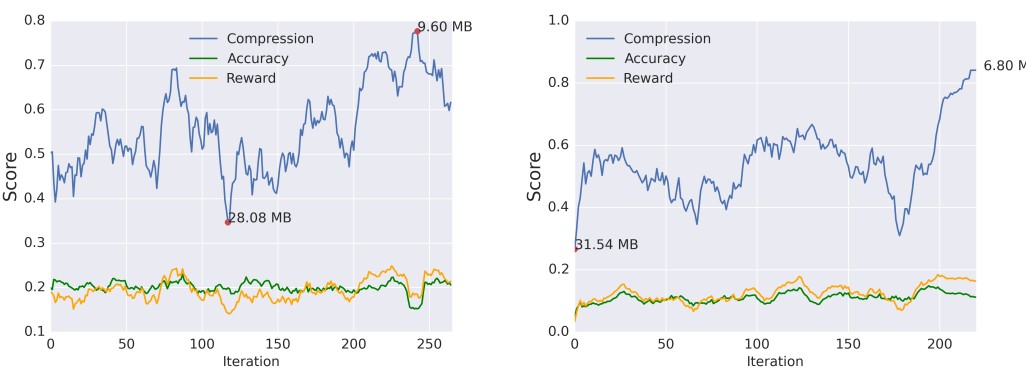

Figure 12: ResNet-18 experiments on **Caltech**, (**Left**: Stage 1, **Right**: Stage 2)

# 10 IMPLEMENTATION DETAILS

The following section contains the implementation details required to replicate the experiments. All of the experiments were implemented in PyTorch with 1 NVIDIA TitanX GPU.

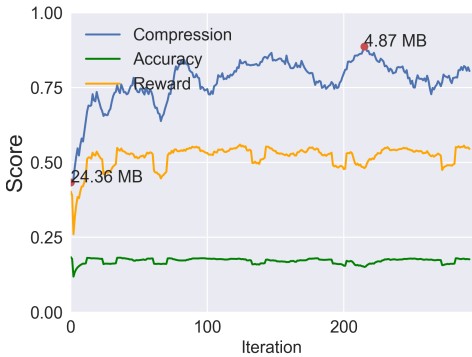

Figure 13: Stage1 ResNet-34 experiments on **ImageNet32x32**

### 10.1 POLICIES

**Removal policy** The removal policy was implemented with 2 hidden layers and 30 hidden units and trained with the Adam optimizer and a learning rate of 0.003. The shrinkage policy was implemented with 2 hidden layers and 50 hidden units and trained with the Adam optimizer and with a learning rate of 0.1. These policies were each trained for at least 100 epochs for each experiment. Batch size of 5 rollouts was used.

### 10.2 TEACHER MODELS

**MNIST** Teacher models for MNIST were trained for 50 epochs with a starting learning rate of 0.01. The learning rate is reduced by a factor of 10 in the 30th epoch. A batch size of 64 was used.
**CIFAR-10/100** Teacher models for CIFAR-10/100 were trained for 150 epochs with a starting learning rate of 0.001. The learning rate is decreased by a factor of 10 in the 80th and 120th epochs. Standard data augmentation with horizontal mirroring (p=0.5), random cropping with padding of 4 pixels and mean subtraction of (0.5, 0.5, 0.5). A batch size of 128 was used.
**SVHN** Teacher models for SVHN were trained for 150 epochs with a starting learning rate of 0.001. The learning rate is decreased by a factor of 10 in the 80th and 120th epochs. Mean subtraction of (0.5, 0.5, 0.5) and a batch size of 128 was used.
**Caltech256** To make the experiments controlled over all datasets the Caltech256 models were trained from scratch. It is to be noted that Caltech256 models are usually initialized with pre-trained ImageNet weights since data is sparse. The training procedure consisted of 50 epochs with an initial learning rate of 0.01. It was reduced to 0.001 after the 50th epoch. Data augmentation such as horizontal flipping and random cropping alongside mean subtraction was used. **ImageNet32x32** The ResNet-34 teacher model for the ImageNet32x32 experiment was trained using a method similar to that described in Chrabaszcz et al. (2017). It was trained for 40 epochs with a starting learning rate of 0.01. The learning rate was reduced by a factor of 5.0 every 10 epochs. Mean subtraction was used with a batch size of 128.

## 11 REWARD DESIGN

In this section we go into greater detail regarding the design of the chosen reward function compared to a naive reward. For our objective of model compression, we want the reward to reflect the following qualitative heuristics.

1. A model with ↑ compression but ↓ accuracy should be penalized more than a model with ↓ compression and ↑ accuracy. Since we do not want to produce highly compressed models which do not perform well on the task, we do not want to let the compression score dominate the reward.

2. The reward function should montonically increase with both compression and accuracy.

## 11.1 NAIVE APPROACH

Defining a naive, symmetrical reward function results in the following failure case. Suppose we define our reward as:

$$R = A * C$$

where $A, C$ are the relative validation accuracy and compression achieved by the student model. Let us consider the following 2 cases:

1. $\uparrow$ accuracy, $\downarrow$ compression. A = 1, C = 0.25
2. $\downarrow$ accuracy, $\uparrow$ compression. A = 0.25, C = 1

In both cases $R = A * C = 0.25$, which we do not want. If we use the reward function defined in the paper we get a reward of 0.25 and 0.4375 for each of the cases, which is closer to our true objective. In our empirical experiments, the non-linear reward outperformed the naive one. Other more complex reward functions that respect the above criteria may also work well.

The visualization of the reward manifold in Figure 14 better illustrates the difference. As observed,

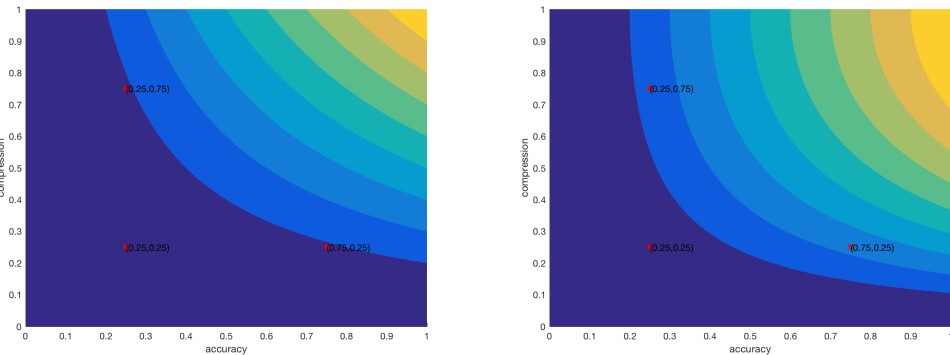

Figure 14: Reward manifold of naive reward vs. our reward

a naive reward function is symmetric while our reward function returns a lower reward for low accuracy, high compression models compared to high accuracy, low compression models. Both functions are monotonically increasing.

## 11.2 DEGENERATE CASES

The following section outlines a few of the cases which are considered degenerate and for which a fixed reward of -1 is assigned.

1. **Empty architecture** - Depending on how it is implemented, the policies could possibly output "remove" actions for each layer during the layer removal stage. In this case, the output would be an empty architecture with no trainable parameters.
2. **Large FC layer** - If too many layers are removed in the feature extraction portion of the convolutional neural network, the size of the feature map before the fully connected layers would be large. In this case, although we have a well defined reward, training the network could be impractical
3. **Specialized architectures** - When dealing with more complex architectures, there may be inter-layer dependencies which impose certain requirements. For example, in a ResNet, the dimensionality of the feature maps at the start and end of each residual block has to match.

## 12 TOTAL TRAINING TIME

To give the reader an approximate estimate of the time taken to train the policies, we have included Table 6 which shows the time taken to train a layer removal policy for 100 iterations. These experiments were done in PyTorch with a single NVIDIA TitanX GPU and an Intel Xeon E5-2660

Table 6: Training time of Layer Removal policy (100 iterations)

| Architecture | Time (hrs) |
|---|---|
| MNIST | |
| VGG-13 | 4 |
| CIFAR-10 | |
| VGG-19 | 17 |
| ResNet-18 | 17 |
| ResNet-34 | 54 |
| SVHN | |
| ResNet-18 | 22 |
| CIFAR-100 | |
| ResNet-18 | 20 |
| ResNet-34 | 55 |
| Caltech256 | |
| ResNet-18 | 175 |
| ImageNet32x32 | |
| ResNet-34 (batch_size=3) | 272 |

CPU. We note that runtime varies based on many factors such as hardware, machine usage and the inherent stochasticity in the approach. The times listed are simply an approximate estimate to how long the method takes on average.

## 13 FUTURE DIRECTIONS

This paper introduces a general method to generate an architecture that optimizes the size-capacity trade-off with respect to a particular task. The current limitation with this method is that we need to train each student model for a few epochs to determine a reward for it. This step can be computationally expensive depending on the dataset. Results from Saxe et al. (2011), Jarrett et al. (2009) and Cox & Pinto (2011) seem to suggest that initializing models with random weights could be an efficient way to evaluate architectures provided the right non-linearities and pooling are used. Another way to provide a better initialization could be to use a hypernetwork which takes the student model architecture as input and produces weights for the model. Other methods that select an informative subset of the training and test dataset to efficiently evaluate the network could also be interesting to explore. Another interesting direction would be to use the pretrained policies for transfer learning on different architecture search problems (apart from compression) to see if any generalizable information about deep architectures is being learned.

