# OpenReview forum: "N2N learning: Network to Network Compression via Policy Gradient Reinforcement Learning"
_ICLR.cc/2018/Conference — Accept (Poster)_

### Official Review · AnonReviewer2 · 2017-11-27
**This paper proposes to use reinforcement learning instead of pre-defined heuristics to determine the structure of the compressed model in the knowledge distillation process.**

**Rating:** 5
**Confidence:** 4

**Review:**

This paper proposes to use reinforcement learning instead of pre-defined heuristics to determine the structure of the compressed model in the knowledge distillation process.

The draft is well-written, and the method is clearly explained. However, I have the following concerns for this draft:

1. The technical contribution is not enough. First, the use of reinforcement learning is quite straightforward. Second, the proposed method seems not significantly different from the architecture search method in [1][2] – their major difference seems to be the use of “remove” instead of “add” when manipulating the parameters. It is unclear whether this difference is substantial, and whether the proposed method is better than the architecture search method.

2. I also have concern with the time efficiency of the proposed method. Reinforcement learning involves multiple rounds of knowledge distillation, and each knowledge distillation is an independent training process that requires many rounds of forward and backward propagations. Therefore, the whole reinforcement learning process seems very time-consuming and difficult to be generalized to big models and large datasets (such as ImageNet). It would be necessary for the authors to make direct discussions on this issue, in order to convince others that their proposed method has practical value.

[1] Zoph, Barret, and Quoc V. Le. "Neural architecture search with reinforcement learning." ICLR (2017).
[2] Baker, Bowen, et al. "Designing Neural Network Architectures using Reinforcement Learning." ICLR (2017).

---

> ### Author Response · Authors · 2017-12-19
> **RE: This paper proposes to use reinforcement learning instead of pre-defined heuristics to determine the structure of the compressed model in the knowledge distillation process.**
>
>
> >>> 2. I also have concern with the time efficiency of the proposed method. Reinforcement learning involves multiple rounds of knowledge distillation, and each knowledge distillation is an independent training process that requires many rounds of forward and backward propagations. Therefore, the whole reinforcement learning process seems very time-consuming and difficult to be generalized to big models and large datasets (such as ImageNet). It would be necessary for the authors to make direct discussions on this issue, in order to convince others that their proposed method has practical value.
>
> The second point is a reasonable criticism, which we have ourselves mentioned and discussed in the appendix (Section 12). Efficiency is a criticism of current architecture search methods in general. Evaluating architectures to obtain a discriminative signal for learning is fundamentally expensive process and is currently an active research topic. Our paper does indeed address this by proposing several improvements over existing architecture search methods.
>
> We define a bounded state space (teacher architecture) and a two-stage policy system in order to reduce the length of rollouts and make credit assignment more efficient (Section 3.2). We also demonstrate generalization experiments which could improve training on larger models (Section 4.6). We think that there are many interesting research directions that can be explored.
>
> To address issues regarding efficiency of our approach more directly, we have run experiments on ImageNet32x32 [3], which we will include in the next revision. We hope that this larger scale experiment directly addresses the concern of running too many iterations of training to find a good compressed architecture. We will also include average runtimes for various networks and datasets in order to give the reader a sense of how long the experiments take. We hope that these updated results would be sufficient to convince the reviewer that the approach is not prohibitive in practice.
>
> [1] Zoph, Barret, and Quoc V. Le. "Neural architecture search with reinforcement learning." ICLR (2017).
> [2] Baker, Bowen, et al. "Designing Neural Network Architectures using Reinforcement Learning." ICLR (2017).
> [3] Chrabaszcz, Patryk et al. “A Downsampled Variant of ImageNet as an Alternative to the CIFAR datasets.” ArXiV (2017).

---

> ### Author Response · Authors · 2017-12-19
> **RE: This paper proposes to use reinforcement learning instead of pre-defined heuristics to determine the structure of the compressed model in the knowledge distillation process.**
>
> >>>1. The technical contribution is not enough. First, the use of reinforcement learning is quite straightforward. Second, the proposed method seems not significantly different from the architecture search method in [1][2] – their major difference seems to be the use of “remove” instead of “add” when manipulating the parameters. It is unclear whether this difference is substantial, and whether the proposed method is better than the architecture search method.
>
> We thank you for your comments and feedback. The second point is helpful to improving our work and we have taken steps to address the comment and improve the paper. The first point however, mischaracterizes our work by stating that “the major difference is ‘remove’ instead of ‘add’” and is not supported by any discussion of the critical details of our approach. It also indiscriminately trivializes the significance of the growing field of model compression. While at a high level, model compression does require removing parameters as opposed to adding parameters, the important research question is *how* parameters should be removed. Our answer to this question is composed of three technical contributions and critical details of the paper: (1) Two-stage policy structure and design of search space (2) Generalization capabilities of learned compression agent (3) Multiobjective reward function and constraints. The criticism of our technical contribution was not supported by any discussion of these contributions -- we hope you can provide an evaluation based on these key details of our work, which we highlight below:
>
> (1) Two-stage policy structure and design of search space: We dedicated Section 3.2 to describing a novel two-stage learning procedure which is critical for learning a better model architecture for the task of model compression. The basic idea is that the architecture search performs a coarse-to-fine search strategy to evaluate large structural changes (i.e. number of layers) before fine tuning each component (i.e., filter size). This not only reduces the computation required to train models in the second stage (since models have been compressed on a macro level in the first stage), but also reduces the dimensionality of the action sequence in both stages, making credit assignment easier. Again, our experiments provide empirical evidence showing that this approach works well. To the best of our knowledge, this is the first work to describe such a strategy and demonstrate its effectiveness. We would like to hear comments that take these technical contributions into account.
>
> (2) Generalization capabilities of our compression networks: Section (4.6) outlines our method for learning generalized policies for a family of networks (e.g., ResNet, VGG). The basic idea is that since many deep learning practitioners typically use a common subset of successful network architectures, it is essential that we can learn policies that can generalize across specific families of teacher networks. We have provided a method for learning such policies and have shown empirically that a single policy can be used for entire family of teacher networks. To the best of our knowledge, this is the first work to show the possibility of such generalization to families of architectures. Furthermore, this is a unique contribution of our method, and one that is not directly applicable to [1, 2] since they build architectures from scratch while we use the teacher model as the initialization. R2 has offered no comments on this result and experiments.
>
> (3) Multiobjective reward function and constraints: Our task is not simply to obtain high statistical performance as in [1, 2], but to achieve compression while maintaining good statistical performance, a competing objective. [1, 2] therefore cannot perform the task of automatic architecture search for compression. Our approach on the other hand, does perform automatic architecture search for compression. We achieve this by introducing model compression specific reward-balancing and constraint satisfaction approaches as detailed in Section 3.3. One could argue that using the approaches of [1, 2] with a modified reward of compression and accuracy could be compared to our approach. However, this approach could result in lengthy models which are small in number of parameters (e.g. too many consecutive ReLUs). If an additional constraint on model length is added to the reward, the optimization procedure becomes harder and it is unclear whether such an approach would have any advantages over our proposed method. Section (11) on reward design covers some of the challenges with designing a reward function for model compression. This important contribution over [1,2] was left unmentioned by R2; we hope they can offer their opinion on this point.

---

### Official Review · AnonReviewer3 · 2017-12-01
**very good paper**

**Rating:** 9
**Confidence:** 4

**Review:**

Summary:
The manuscript introduces a principled way of network to network compression, which uses policy gradients for optimizing two policies which compress a strong teacher into a strong but smaller student model. The first policy, specialized on architecture selection, iteratively removes layers, starting with architecture of the teacher model. After the first policy is finished, the second policy reduces the size of each layer by iteratively outputting shrinkage ratios for hyperparameters such as kernel size or padding. This organization of the action space, together with a smart reward design achieves impressive compression results, given that this approach automates tedious architecture selection. The reward design favors low compression/high accuracy over high compression/low performance while the reward still monotonically increases with both compression and accuracy. As a bonus, the authors also demonstrate how to include hard constraints such as parameter count limitations into the reward model and show that policies trained on small teachers generalize to larger teacher models.

Review:
The manuscript describes the proposed algorithm in great detail and the description is easy to follow. The experimental analysis of the approach is very convincing and confirms the author’s claims.
Using the teacher network as starting point for the architecture search is a good choice, as initialization strategies are a critical component in knowledge distillation. I am looking forward to seeing work on the research goals outlined in the Future Directions section.

A few questions/comments:
1) I understand that L_{1,2} in Algorithm 1 correspond to the number of layers in the network, but what do N_{1,2} correspond to? Are these multiple rollouts of the policies? If so, shouldn’t the parameter update theta_{{shrink,remove},i} be outside the loop over N and apply the average over rollouts according to Equation (2)? I think I might have missed something here.
2) Minor: some of the citations are a bit awkward, e.g. on page 7: “algorithm from Williams Williams (1992). I would use the \citet command from natbib for such citations and \citep for parenthesized citations, e.g. “... incorporate dark knowledge (Hinton et al., 2015)” or “The MNIST (LeCun et al., 1998) dataset...”
3) In Section 4.6 (the transfer learning experiment), it would be interesting to compare the performance measures for different numbers of policy update iterations.
4) Appendix: Section 8 states “Below are the results”, but the figure landed on the next page. I would either try to force the figures to be output at that position (not in or after Section 9) or write "Figures X-Y show the results". Also in Section 11, Figure 13 should be referenced with the \ref command
5) Just to get a rough idea of training time: Could you share how long some of the experiments took with the setup you described (using 4 TitanX GPUs)?
6) Did you use data augmentation for both teacher and student models in the CIFAR10/100 and Caltech256 experiments?
7) What is the threshold you used to decide if the size of the FC layer input yields a degenerate solution?

Overall, this manuscript is a submission of exceptional quality and if minor details of the experimental setup are added to the manuscript, I would consider giving it the full score.

---

> ### Author Response · Authors · 2017-12-19
> **RE: very good paper**
>
> We thank R3 for their thorough and detailed review of the paper. We have included our responses to the questions below and made the relevant changes to our paper where required.
>
> >>> 1) I understand that L_{1,2} in Algorithm 1 correspond to the number of layers in the network, but what do N_{1,2} correspond to? Are these multiple rollouts of the policies? If so, shouldn’t the parameter update theta_{{shrink,remove},i} be outside the loop over N and apply the average over rollouts according to Equation (2)? I think I might have missed something here.
> #1. The rollouts were omitted in order to simplify the presentation of the algorithm. N refers to the number of total iterations (or policy updates) for which the policy is trained.
> >>> 2) Minor: some of the citations are a bit awkward, e.g. on page 7: “algorithm from Williams Williams (1992). I would use the \citet command from natbib for such citations and \citep for parenthesized citations, e.g. “... incorporate dark knowledge (Hinton et al., 2015)” or “The MNIST (LeCun et al., 1998) dataset...”
> #2. Thank you for this suggestion, we have fixed the citations in the new revision.
> >>> 3) In Section 4.6 (the transfer learning experiment), it would be interesting to compare the performance measures for different numbers of policy update iterations.
> #3. Figure 10 in the appendix shows the plots over multiple policy update iterations when the pre-trained policies are used.
> >>> 4) Appendix: Section 8 states “Below are the results”, but the figure landed on the next page. I would either try to force the figures to be output at that position (not in or after Section 9) or write "Figures X-Y show the results". Also in Section 11, Figure 13 should be referenced with the \ref command
> #4. We have fixed this in the new revision.
> >>> 5) Just to get a rough idea of training time: Could you share how long some of the experiments took with the setup you described (using 4 TitanX GPUs)?
> #5. We have added a new section to the appendix in the revision providing details on the runtime of the experiments. In general, the shortest experiment (VGG-13/MNIST) took about 4 hours, while the longest experiment (ResNet34/ImageNet32x32) took about 272 hours in total. The experiments actually only used a single TitanX GPU. We have updated the paper to reflect this.
> >>> 6) Did you use data augmentation for both teacher and student models in the CIFAR10/100 and Caltech256 experiments?
> #6. Yes we used standard data augmentation techniques. This is discussed in Section 10.2 of the paper.
> >>> 7) What is the threshold you used to decide if the size of the FC layer input yields a degenerate solution?
> #7. We say the network is degenerate if its size exceeds that of the teacher or if the size of the FC layer is greater than 50,000.

---

### Official Review · AnonReviewer1 · 2017-12-03
**Reinforcement learning for estimating the structure of the compressed model in the knowledge distillation process**

**Rating:** 4
**Confidence:** 4

**Review:**

On the positive side the paper is well written and the problem is interesting.

On the negative side there is very limited innovation in the techniques proposed, that are indeed small variations of existing methods.

---

> ### Author Response · Authors · 2017-12-19
> **RE: Reinforcement learning for estimating the structure of the compressed model in the knowledge distillation process**
>
> We thank you for the review. We would appreciate if the review contained a more concrete technical discussion of the work instead of unsupported negative statements. We hope that the reviewer appreciates that we have put in substantial work into this paper and is willing to continue this discussion in a more meaningful manner.
>
> This review appears to repeat the criticism brought up by R2, which we strongly disagree with. We have explained our stance and provided supporting details in the response below.
>
> Our answer to this question is composed of three technical contributions and critical details of the paper: (1) Two-stage policy structure and design of search space (2) Generalization capabilities of learned compression agent (3) Multiobjective reward function and constraints. The criticism of our technical contribution was not supported by any discussion of these contributions -- we hope you can provide an evaluation based on these key details of our work, which we highlight below:
>
> (1) Two-stage policy structure and design of search space: We dedicated Section 3.2 to describing a novel two-stage learning procedure which is critical for learning a better model architecture for the task of model compression. The basic idea is that the architecture search performs a coarse-to-fine search strategy to evaluate large structural changes (i.e. number of layers) before fine tuning each component (i.e., filter size). This not only reduces the computation required to train models in the second stage (since models have been compressed on a macro level in the first stage), but also reduces the dimensionality of the action sequence in both stages, making credit assignment easier. Again, our experiments provide empirical evidence showing that this approach works well. To the best of our knowledge, this is the first work to describe such a strategy and demonstrate its effectiveness. We would like to hear comments that take these technical contributions into account.
>
> (2) Generalization capabilities of our compression networks: Section (4.6) outlines our method for learning generalized policies for a family of networks (e.g., ResNet, VGG). The basic idea is that since many deep learning practitioners typically use a common subset of successful network architectures, it is essential that we can learn policies that can generalize across specific families of teacher networks. We have provided a method for learning such policies and have shown empirically that a single policy can be used for entire family of teacher networks. To the best of our knowledge, this is the first work to show the possibility of such generalization to families of architectures. R1 has offered no comments on this result and experiments.
>
> (3) Multiobjective reward function and constraints: Our task is not simply to obtain high performance as in [1, 2], but to achieve compression while maintaining good performance, a competing objective. [1, 2] therefore cannot perform the task of automatic architecture search for compression. Our approach on the other hand, does perform automatic architecture search for compression. We achieve this by introducing model compression specific reward-balancing and constraint satisfaction approaches as detailed in Section 3.3. One could argue that using the approaches of [1, 2] with a modified reward of compression and accuracy could be compared to our approach. However, this approach could result in lengthy models which are small in number of parameters (e.g. too many consecutive ReLUs). If an additional constraint on model length is added to the reward, the optimization procedure becomes harder and it is unclear whether such an approach would have any advantages over our proposed method. Section (11) on reward design covers some of the challenges with designing a reward function for model compression. This important contribution over [1,2] was left unmentioned by R1; we hope they can offer their opinion on this point.
>
> [1] Zoph, Barret, and Quoc V. Le. "Neural architecture search with reinforcement learning." ICLR (2017).
> [2] Baker, Bowen, et al. "Designing Neural Network Architectures using Reinforcement Learning." ICLR (2017).

---

### Decision · Program_Chairs · 2018-01-29
**ICLR 2018 Conference Acceptance Decision**

**Decision:**

Accept (Poster)

**Comment:**

This is a meta-learning approach to model compression which trains 2 policies using RL to reduce the capacity (computational cost) of a trained network while maintaining performance, such that it can be effectively transferred to a smaller student network. The approach has similarities to recently proposed methods for architecture search, but is significantly different. The paper is well written and the experiments are clear and convincing. One of the reviews was unacceptable; I am not considering it (R1).